# The CDK Pef1 and protein phosphatase 4 oppose each other for regulating cohesin binding to fission yeast chromosomes

Adrien Birot[1†], Marta Tormos-Pérez[1†], Sabine Vaur[1†], Amélie Feytout[1], Julien Jaegy[1], Dácil Alonso Gil[1], Stéphanie Vazquez[1], Karl Ekwall[2], Jean-Paul Javerzat[1]*

[1]Institut de Biochimie et Génétique Cellulaires, UMR 5095 CNRS - Université de Bordeaux, Bordeaux, France; [2]Department of Biosciences and Nutrition, Karolinska Institutet, Huddinge, Sweden

**Abstract** Cohesin has essential roles in chromosome structure, segregation and repair. Cohesin binding to chromosomes is catalyzed by the cohesin loader, Mis4 in fission yeast. How cells fine tune cohesin deposition is largely unknown. Here, we provide evidence that Mis4 activity is regulated by phosphorylation of its cohesin substrate. A genetic screen for negative regulators of Mis4 yielded a CDK called Pef1, whose closest human homologue is CDK5. Inhibition of Pef1 kinase activity rescued cohesin loader deficiencies. In an otherwise wild-type background, Pef1 ablation stimulated cohesin binding to its regular sites along chromosomes while ablating Protein Phosphatase 4 had the opposite effect. Pef1 and PP4 control the phosphorylation state of the cohesin kleisin Rad21. The CDK phosphorylates Rad21 on Threonine 262. Pef1 ablation, non-phosphorylatable Rad21-T262 or mutations within a Rad21 binding domain of Mis4 alleviated the effect of PP4 deficiency. Such a CDK/PP4-based regulation of cohesin loader activity could provide an efficient mechanism for translating cellular cues into a fast and accurate cohesin response.

**\*For correspondence:**
jpaul.javerzat@ibgc.cnrs.fr

[†]These authors contributed equally to this work

**Competing interests:** The authors declare that no competing interests exist.

## Introduction

Cohesin is a central player in chromosome biology. Defects in the cohesin pathway are linked to human pathologies such as sterility, cancer and severe developmental disorders (*Watrin et al., 2016*; *Cheng and Liu, 2017*; *De Koninck and Losada, 2016*). Cohesin is an ATPase-driven molecular machine able to tether DNA by topological entrapment (*Ivanov and Nasmyth, 2005*; *Haering et al., 2008*; *Gligoris et al., 2014*). The core cohesin complex is made of two long Structural Maintenance of Chromosome (SMC) proteins (Psm1 and Psm3 in fission yeast) whose ATPase heads are bridged by a kleisin subunit called Scc1/Mcd1/Rad21. The cohesin complex ensures proper chromosome segregation by holding sister chromatids together from DNA replication and until their segregation at anaphase onset. Besides chromosome segregation, cohesin is essential for DNA repair and the formation of DNA loops that shape chromosome architecture and impinge on gene regulation (*Makrantoni and Marston, 2018*; *van Ruiten and Rowland, 2018*; *Arzate-Mejía et al., 2018*). Cohesin is therefore central to many biological processes, emphasizing the importance of understanding its regulation.

Cohesin loading onto DNA requires ATP hydrolysis by cohesin, which is thought to trigger transient opening of the ring and DNA entrapment (*Weitzer et al., 2003*; *Gruber et al., 2006*; *Arumugam et al., 2003*; *Murayama and Uhlmann, 2015*; *Srinivasan et al., 2018*). The reaction is stimulated by a separate complex, the cohesin loader, made of Scc2-Scc4 in yeast, NIPBL-MAU2 in human, Mis4-Ssl3 in fission yeast (*Ciosk et al., 2000*; *Watrin et al., 2006*; *Bernard et al., 2006*). Structural studies indicate that Scc2 consists of an N-terminal globular domain that binds Scc4

followed by helical repeats that fold into a hook-shaped structure (*Chao et al., 2015*; *Kikuchi et al., 2016*). Mis4 alone has DNA-binding activity and the C terminal hook domain is sufficient for stimulating DNA capture by cohesin in vitro (*Chao et al., 2015*; *Murayama and Uhlmann, 2014*). Mis4 makes multiple contacts with cohesin which may help cohesin conformational changes required for DNA capture (*Kikuchi et al., 2016*; *Murayama and Uhlmann, 2014*). Although dispensable in vitro, Ssl3 is essential for cohesin binding to chromosomes (*Bernard et al., 2006*; *Murayama and Uhlmann, 2014*). Scc4 wraps around Scc2 N terminus and is thought to recognize and bind chromatin receptors thereby directing cohesin loading to specific locations (*Chao et al., 2015*; *Murayama and Uhlmann, 2014*; *Fernius et al., 2013*; *Takahashi et al., 2008*; *Hinshaw et al., 2017*; *Hinshaw et al., 2015*; *Muñoz et al., 2019*).

Once loaded onto chromosomes, cohesin can either remain bound or removed. Wpl1 promotes cohesin release in a reaction requiring Pds5 and Psc3. All three proteins bind Rad21 and Wpl1 promotes DNA release by weakening the Rad21-Smc3 interface (*Gandhi et al., 2006*; *Kueng et al., 2006*; *Rowland et al., 2009*; *Lopez-Serra et al., 2013*; *Chan et al., 2012*; *Beckouët et al., 2016*; *Huis in 't Veld et al., 2014*). DNA release is counteracted by a cohesin acetyl-transferase (Eso1 in fission yeast) that acetylates Smc3 during S phase (*Rowland et al., 2009*; *Rolef Ben-Shahar et al., 2008*; *Zhang et al., 2008*; *Feytout et al., 2011*; *Unal et al., 2008*).

As opposed to sister-chromatid cohesion, cohesin loops may not require topological entrapment of DNA (*Srinivasan et al., 2018*). Cohesin may capture small loops of DNA and then extrude them in a processive manner. The formation of cohesin loops is dependent on the loading complex and reciprocally loops are de-stabilized by Wpl1 (*Tedeschi et al., 2013*; *Haarhuis et al., 2017*). Topological versus non-topological DNA capture may be achieved by modulating the catalytic activity of the loader and concerted transcriptional responses may involve a local and temporal control of loading and unloading activities. How cells orchestrate cohesin functions is largely unknown.

Intriguingly, the kleisin subunit of cohesin is targeted by multiple phosphorylation events. In fission yeast, Rad21 shows multiple phospho-isoforms whose relative abundance fluctuates along the cell cycle (*Adachi et al., 2008*). Our recent work showed that Protein Phosphatase 4 (PP4) controls the phosphorylation status of Rad21 and modulates Wpl1 activity (*Birot et al., 2017*), leading to the idea that cohesin functions could be spatially and temporally fine-tuned by altering the balance between kinase and phosphatase activities.

Here, we report on the control of cohesin deposition by the opposite activities of the Pef1 CDK and PP4. Pef1 was first described as a PSTAIRE-related protein in fission yeast (*Tournier et al., 1997*). The CDK has three known cyclin partners called Pas1, Psl1 and Clg1 and was reported to facilitate the G1 to S phase transition and to regulate life span (*Chen et al., 2013*; *Tanaka and Okayama, 2000*). Its closest human homolog, CDK5, is involved in a myriad of cellular functions and pathologies, from neurodegenerative diseases to multiple solid and hematological cancers (*Lenjisa et al., 2017*). We identified *pef1* in a genetic screen for mutants able to rescue the cohesin loader mutant *mis4-367*. Pef1 ablation or inhibition of its kinase activity increased cohesin deposition and rescued sister-chromatid cohesion defects of the *mis4* mutant. In otherwise wild-type cells, Pef1 ablation increased the binding of both cohesin and its loader to their regular sites along chromosomes. Genetic analyses indicated that Pef1 acts through the phosphorylation of multiple targets. We identified one of these within the kleisin Rad21. Specifically, the Pef1/Psl1 complex phosphorylates Rad21 on T262 and preventing this phosphorylation event recapitulates in part the effects of Pef1 ablation. PP4 had the opposite effect. Its ablation lead to hyper-phosphorylated Rad21 and reduced cohesin deposition which is alleviated by Pef1 ablation or Rad21-T262A. Hence, phosphorylation of the kleisin negatively regulates cohesin loading, possibly by lowering the activity of the cohesin loader. Further supporting this notion, a genetic screen identified compensatory mutations that cluster within the catalytic domain of Mis4, in a previously described Rad21-binding region. Such a phosphorylation-based control may provide a fast, accurate and reversible way for regulating cohesin functions in response to cellular cues.

## Results

### Inhibition of Pef1 kinase activity in *mis4-367* cells increases cohesin binding to DNA in S phase and improves chromosome segregation during mitosis

The *mis4-367* allele encodes Mis4[G1487D]. This single amino acid change is located within the last HEAT repeat of the C-terminal catalytic domain (*Figure 1A*), rendering the strain thermosensitive for growth (ts). To identify putative regulators of Mis4, we made a genetic screen for suppressors of the ts phenotype, the rationale being that loss of a negative regulator should upregulate residual Mis4[G1487D] activity and restore growth at the restrictive temperature. Eleven mutants were isolated that distributed into four linkage groups. Genetic mapping and tiling array hybridization were used to identify the mutated locus in group 1. A single base substitution was found within the *pef1* coding sequence. The amino acid change (N146S) is located within the catalytic site of the kinase suggesting the kinase activity was involved. Accordingly, deletion of the *pef1* gene or inhibition of Pef1 kinase activity using an analog-sensitive allele (*pef1-as*) suppressed the ts growth defect of *mis4-367* (*Figure 1B*). Likewise, *pef1Δ* showed a suppressor effect towards the strong ts allele *mis4-242* (*Takahashi et al., 1994*) and efficiently suppressed *ssl3-29* (*Figure 1—figure supplement 1*), a ts mutant of *ssl3* (*Bernard et al., 2006*). The deletion of *pef1* even allowed cell survival in the complete absence of the *ssl3* gene, although colonies were tiny and grew very slowly (*Figure 1—figure supplement 1*). By contrast *pef1Δ* showed a negative genetic interaction with *eso1-H17* (*Tanaka et al., 2000*) indicating that *pef1* displays distinct genetic interactions with components of the cohesin pathway (*Figure 1—figure supplement 1*). Deletion of *pef1* did not allow cell survival in the complete absence of the *mis4* gene (*Figure 1—figure supplement 1*), indicating that *pef1*-mediated suppression required residual Mis4 activity. Altogether, these genetic data suggested that *pef1* deletion may upregulate Mis4. The corollary being that the CDK may act as a negative regulator of Mis4.

We first aimed at characterizing the suppression of Mis4[G1487D] phenotypes by *pef1* mutants. Thermosensitive mutants of the cohesin loading complex fail to properly establish sister-chromatid cohesion during S phase (*Ciosk et al., 2000*; *Bernard et al., 2006*; *Furuya et al., 1998*) and consequently display a high frequency of aberrant mitoses in which sister chromatids lag along the spindle during anaphase. After one complete cell cycle at the restrictive temperature *mis4-367* cells displayed a high frequency of aberrant anaphases, a defect which was efficiently rescued by the deletion of the *pef1* gene. The chemical inhibition of Pef1-as had a similar effect, confirming that Pef1 acts through its kinase activity (*Figure 1C*).

The cohesin loading complex performs its essential function during G1/S (*Ciosk et al., 2000*; *Bernard et al., 2006*; *Furuya et al., 1998*). Accordingly, chromosome segregation was efficiently restored at 36.5℃ when Pef1-as was inhibited before but not after S phase onset (*Figure 1D*). Finally, sister-chromatid cohesion was monitored by FISH, using a probe located close to the centromere of chromosome 2 (*Figure 1E*). The inhibition of Pef1 kinase activity significantly reduced the frequency of separated FISH signals, consistent with improved sister chromatid cohesion.

The failure to establish cohesion during S phase may be due to poor cohesin loading. To see whether the inhibition of Pef1 kinase activity would improve cohesin binding to chromosomes at the time of cohesion establishment, *mis4-367 pef1-as* cells were cultured in the presence of 1-NA-PP1 or solvent alone (DMSO) and arrested in S phase at the restrictive temperature using hydroxyurea (*Figure 2A* and *Figure 2—figure supplement 1*). The amount of chromatin-bound Rad21 was monitored by Chromatin Immunoprecipitation (ChIP) at known Cohesin Associated Regions (CARs) along the arms and centromere of chromosome 2, the rDNA gene cluster on chromosome 3 and the chromosome 1 right telomere (*Figure 2B*) (*Birot et al., 2017*; *Schmidt et al., 2009*). Within the centromere, Rad21 binding was examined at the central core (*cc2*) which is the site of kinetochore assembly, within the *imr* and *dg* repeats that flank the central core on either side and at tRNA-rich domains that delineate the centromere. The peri-centromere repeats are bound by Heterochromatin Protein 1 (Swi6 in fission yeast) that recruits cohesin, thus providing robust sister chromatid cohesion at centromeres (*Bernard et al., 2001*; *Nonaka et al., 2002*).

As expected, Rad21 binding was reduced at all sites examined in the *mis4-367* mutant when compared to wild-type (*Figure 2—figure supplement 1*). To highlight the effect of Pef1-as inhibition, the ratios 1-NA-PP1/DMSO were calculated for each chromosomal site (*Figure 2C*). This shows that

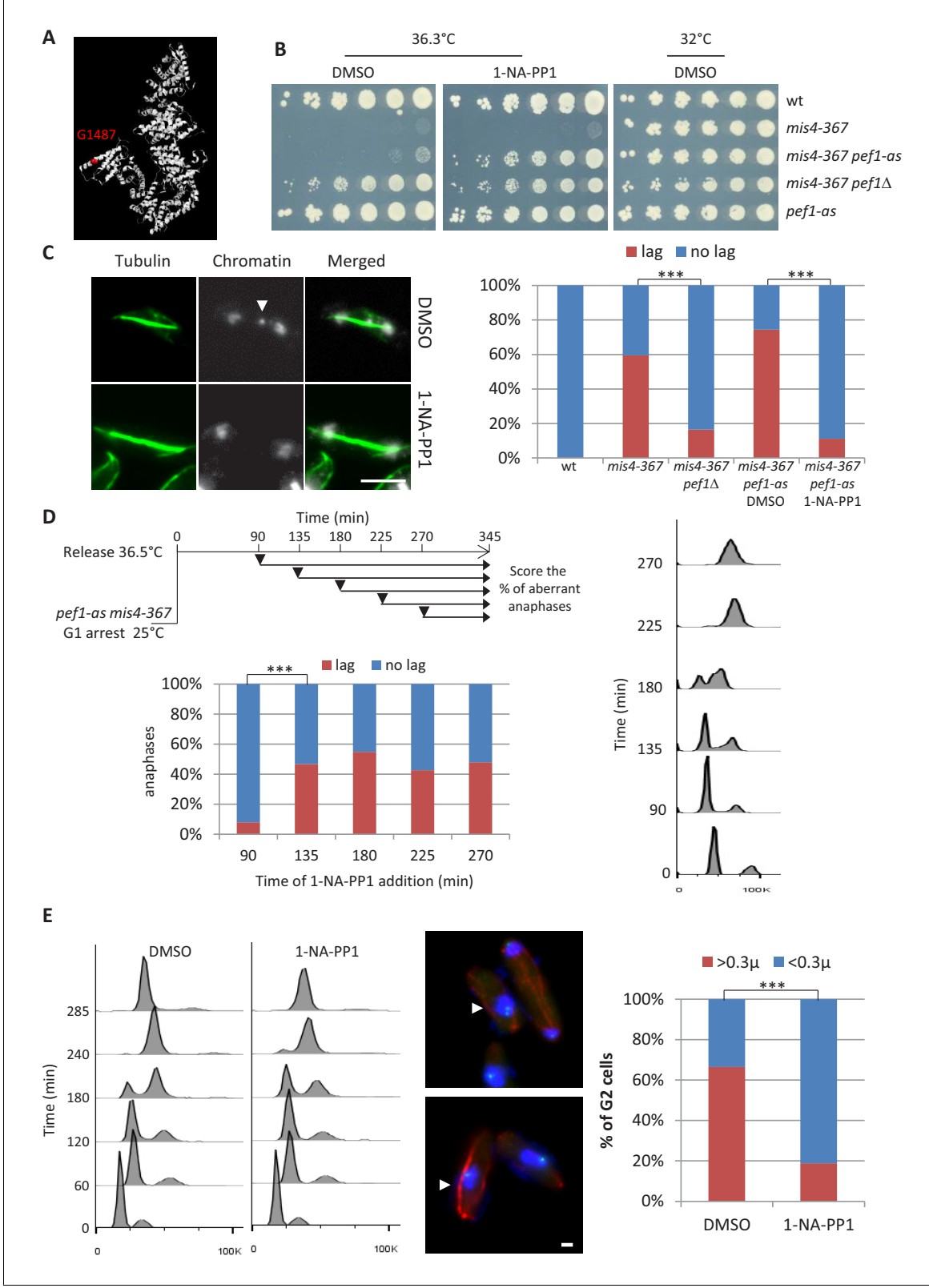

**Figure 1.** Inhibition of Pef1 kinase activity suppressed Mis4$^{G1487D}$ cohesion and chromosome segregation defects. (**A**) The *mis4-367* allele results in a G1487D substitution within the last HEAT repeat of Mis4. (**B**) Cell growth assay showing that inhibition of Pef1 kinase activity suppresses *mis4-367*-thermosensitive growth phenotype. (**C**) Inhibition of Pef1 kinase activity suppresses *mis4-367* chromosome segregation defects. Cells were cultured at 36.5°C for a complete cell cycle. Lagging chromatids appear as DAPI-stained material (arrow) along the anaphase spindle (tubulin staining in green). *Figure 1 continued on next page*

*Figure 1 continued*

Bar = 5 µm. ***p<0.0001 two-sided Fisher's exact test (*Figure 1—source data 1*). (D) Pef1 inhibition must occur before S phase onset to rescue *mis4-367* chromosome segregation. Cells were arrested in G1 by nitrogen starvation, released into the cell cycle at 36.5°C and 1-NA-PP1 added at the indicated time points (arrows). Cell cycle progression was followed by measurement of DNA content. Anaphase cells with lagging chromatids were scored at the 345 min time point. ***p<0.0001 two-sided Fisher's exact test (*Figure 1—source data 1*). (E) Pef1 kinase inhibition improved sister-chromatid cohesion. Cells were arrested in G1 by nitrogen starvation, released into the cell cycle at 36.5°C with or without 1-NA-PP1. Cells were harvested after DNA replication (285 min) and processed for FISH using a centromere two-linked probe. Distance between FISH signals was measured in G2 cells, as judged by DNA content and the interphase array of microtubules. Bar = 1 µm. ***p<0.0001 two-sided Fisher's exact test (*Figure 1—source data 1*).

The online version of this article includes the following source data and figure supplement(s) for figure 1:

**Source data 1.** Statistical tests.
**Figure supplement 1.** pef1 genetic interactions.

Rad21 binding was significantly increased at most CARs in the presence of 1-NA-PP1 and in a *pef1-as*-dependent manner.

When compared to wild-type levels (*Figure 2D*), cohesin binding was significantly restored at two chromosome arm sites (1806, 2898), at the rDNA Non Transcribed Spacer (NTS) to ~60% wild-type levels and ~50% at the telomere site (Tel1-R). Within the centromere, Rad21 binding was back to ~50% wild-type levels within the outer repeats (*imr2-L* and *dg2-R*), consistent with improved sister-chromatid cohesion as seen with the cen2 FISH assay (*Figure 1E*). The suppression of *mis4-367* growth defect by *pef1Δ* was strongly dependent on a functional *swi6* gene (*Figure 2E*), suggesting that the enhancement of cohesin binding within heterochromatin domains is a key determinant of Pef1-mediated suppression.

The total amount of chromatin bound Rad21 per nucleus, as assayed by nuclear spreads (*Figure 2—figure supplement 1*), remained largely unchanged. This may indicate that the global increase remains below the sensitivity of this assay or that Pef1 inhibition primarily affects cohesin distribution along chromosomes.

The establishment of sister chromatid cohesion in S phase is accompanied by acetylation of the cohesin subunit Psm3 (*Feytout et al., 2011*). At 25°C, Psm3 K106 acetylation in *mis4-367* cycling cells was similar to wild-type (*Figure 2—figure supplement 1*). By contrast, the level dropped in S-phase-arrested *mis4-367* cells at 35.5°C. Psm3 K106 acetylation was increased ~4 fold (*Figure 2—figure supplement 1*) when Pef1-as was inhibited. The level of acetylated cohesin remained however very low when compared to wild-type suggesting the increase may not be significant. Alternatively, such an apparent marginal increase may contribute to the observed improvement of sister-chromatid cohesion.

From this set of experiments, we conclude that the inhibition of Pef1 kinase activity enhanced cohesin binding to CARs in Mis4 deficient cells and improved the establishment of sister chromatid cohesion as well as chromosome segregation.

In fission yeast, a small fraction of cohesin may dissociate from chromatin during early mitosis, another is cleaved by separase at anaphase onset while the bulk may remain bound to chromosomes (*Schmidt et al., 2009*; *Tomonaga et al., 2000*). Pef1 inhibition may rescue *mis4-367* by acting on cohesin from the previous cell cycle and/or may stimulate de novo cohesin loading. The latter possibility was investigated by inducing the expression of an ectopic FLAG-tagged *rad21* construct in G1 cells. Growth assays indicated that ectopically expressed *rad21-FLAG* was functional as it allowed cell division in the absence of the endogenous *rad21* gene and *pef1Δ* suppressed the *mis4-367* ts growth defect under these conditions (*Figure 3A*). To see whether Pef1 inhibition would rescue *mis4-367* and allow neo-synthesized Rad21-FLAG to bind chromatin, cells were arrested in G1 at 36.5°C and Rad21-FLAG was induced with or without Pef1-as inhibition (*Figure 3BC*). Rad21-FLAG binding to chromatin was monitored by cell fractionation (*Figure 3DE*). As expected, neo-synthesized Rad21-FLAG was poorly associated with the chromatin fraction when *pef1-as* was not inhibited. By contrast, chromatin-bound Rad21-FLAG was increased upon inhibition of the CDK (*Figure 3D and F*). Hence, the inhibition of Pef1 kinase activity allowed neo-synthesized Rad21 binding to chromatin in the context of crippled cohesin loader activity. The most straightforward interpretation is that Pef1 inhibition stimulated the residual activity of the crippled cohesin loader. Alternatively, enhanced cohesin binding to DNA may result from reduced unloading. Two mechanisms have been

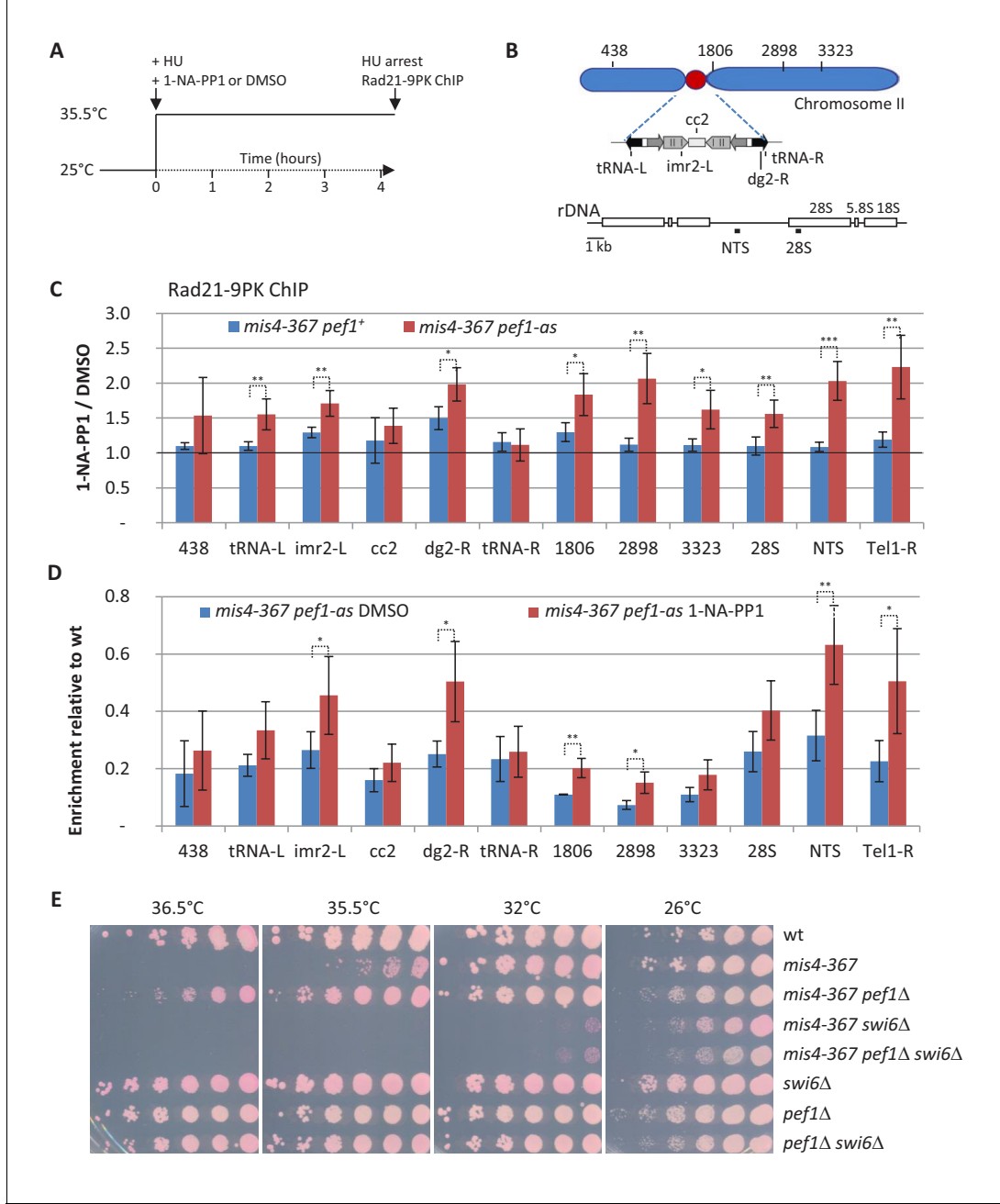

**Figure 2.** Inhibition of the CDK Pef1 in *mis4-367* increased Rad21 binding to S phase chromosomes. (**A**) Scheme of the experiment. Hydroxyurea (HU) was added to 12 mM at the time of the temperature shift along with 1-NA-PP1 or solvent alone (DMSO). Cells were collected after 4.25 hr. The S phase arrest was confirmed by DNA content analysis (*Figure 2—figure supplement 1*). (**B**) Schematics showing the loci analyzed by ChIP-qPCR. (**C**) The effect of 1-NA-PP1 treatment on Pef1-as is shown by the ratio 1-NA-PP1/DMSO (red) for each site analyzed. The ratios in a *pef1*[+] background (blue) estimate the off target effects of the inhibitor. Ratios were calculated from the ChIP data shown in *Figure 2—figure supplement 1*. Bars indicate mean ± SD from four ratios. ***p≤0.001, **p≤0.01, *p≤0.05, by two-tailed, unpaired t-test with 95% confidence interval (*Figure 2—source data 1*). (**D**) Rad21 binding relative to wild-type. The ratios highlight the recovery of Rad21 binding upon inhibition of the CDK relative to wild-type levels. Ratios were calculated from the data shown in *Figure 2—figure supplement 1*. Bars indicate mean ± SD from four ratios. **p≤0.01, *p≤0.05, by two-tailed, unpaired t-test with 95% confidence interval (*Figure 2—source data 1*). (**E**) Cell growth assay showing that *pef1Δ* does not suppress *mis4-367* thermosensitive phenotype in the absence of the *swi6* gene.

The online version of this article includes the following source data and figure supplement(s) for figure 2:

**Source data 1.** Raw ChIP data and t-tests.

**Figure supplement 1.** DNA content analysis, raw ChIP data, Psm3 acetylation and nuclear spreads from HU-arrested cells.

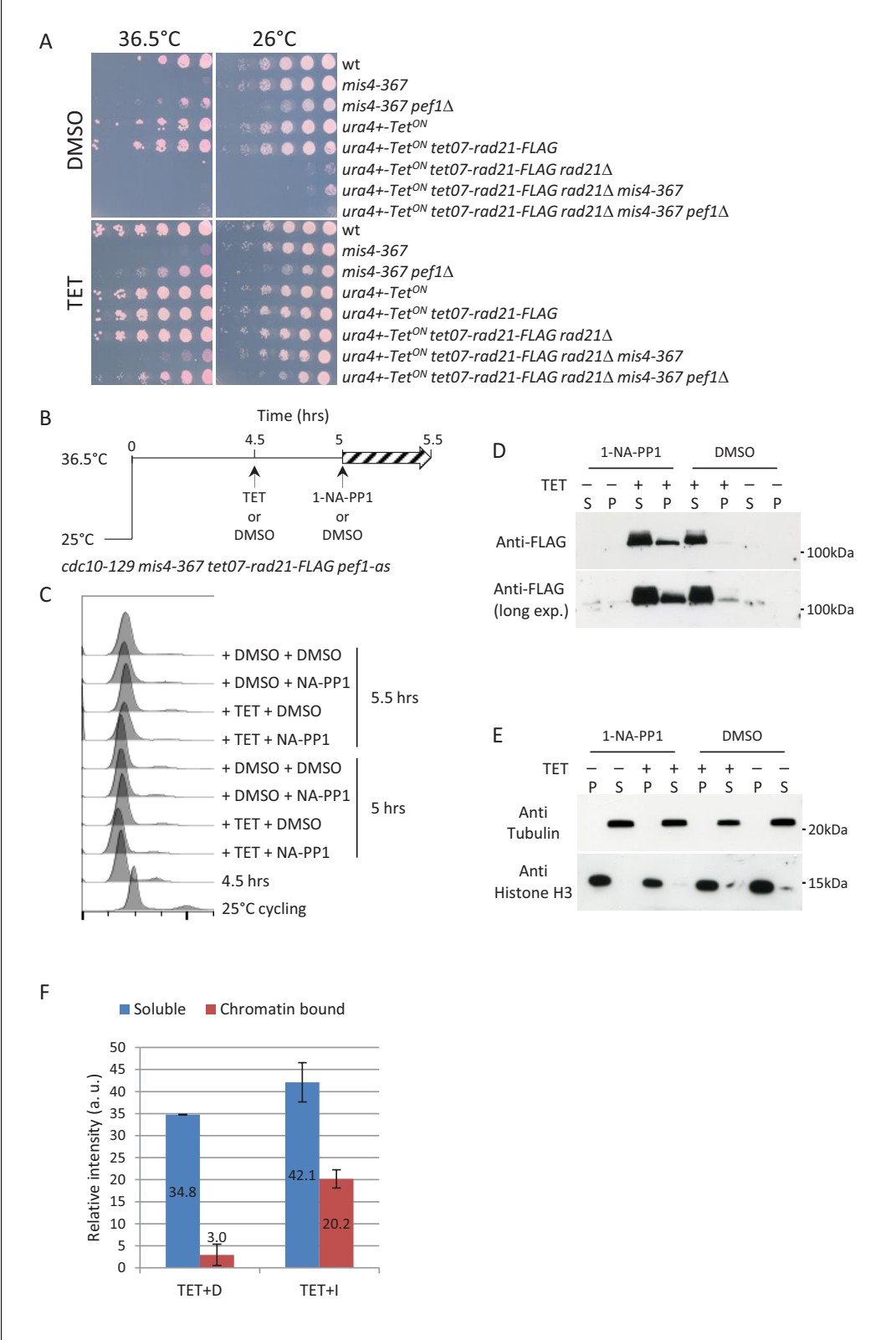

**Figure 3.** Inhibition of Pef1 kinase in G1-arrested *mis4-367* cells allows neo-synthesized Rad21 to bind chromatin. (**A**) The tetracycline (TET) inducible *tet07-rad21-FLAG* construct can substitute for the endogenous *rad21* gene. The last two lanes show that *pef1Δ* suppresses *mis4-367* ts phenotype when *tet07-rad21-FLAG* is the sole source of Rad21. (**B**) Experimental scheme. Cells cultured in EMM2 medium were arrested in G1 by the *cdc10-129* mutation. After 4.5 hr *tet07-rad21-FLAG* was induced by the addition of TET or left un-induced (DMSO). Pef1-as was inhibited 30 min later and samples

*Figure 3 continued on next page*

*Figure 3 continued*

collected after 30 min. (C) DNA content analysis. (D) Western blot analysis of Rad21-FLAG in the chromatin (P) and soluble (S) fractions. (E) Fractionation controls. Anti-tubulin and anti-Histone H3 antibodies were used as markers for the soluble (S) and chromatin (P) fractions, respectively. (F) Rad21-FLAG signals were quantified for the TET samples from the long and short exposure blots shown in (D). The bars represent the mean relative band intensities + / - SD.

The online version of this article includes the following figure supplement(s) for figure 3:

**Figure supplement 1.** Pef1 acts independently from the Psm3/Rad21 interface.

described for unloading cohesin from G1 chromosomes, both involving the opening of the Smc3/Kleisin interface. One is dependent on Wpl1 while the other is not and was recently reported in budding yeast to be inhibited by Scc2 (*Srinivasan et al., 2019*). If *pef1Δ* suppressed *mis4-367* by inhibiting such an unloading mechanism, the artificial closure of the Psm3/Rad21 interface should suppress *mis4-367*. As originally reported in budding yeast (*Chan et al., 2012*), a *psm3-rad21* gene fusion efficiently bypassed the requirement for the Eso1 acetyl-transferase (*Figure 3—figure supplement 1*) but did not restore and even enhanced the temperature growth defect of *mis4-367* (*Figure 3—figure supplement 1*). Importantly, *pef1Δ* still showed a suppressor activity in this genetic setup, indicating that Pef1 acts independently from the Psm3/Rad21 interface. Likewise the deletion of *wpl1* had little effect, if any (*Figure 3—figure supplement 1*). These genetic data argue against a Pef1 mediated control of the Smc3/Rad21 interface. We therefore favor the conclusion that Pef1 inhibition rescued cohesin loader deficiency by increasing its residual activity.

## Chromatin binding of cohesin and its loader Mis4 are regulated by Pef1

The above data suggest that the residual cohesin loading activity in *mis4-367* is enhanced when Pef1 kinase is inhibited, implying that the CDK may function as a negative regulator of Mis4. We addressed this question by looking at the effect of Pef1 ablation on Rad21 and Mis4 binding to chromosomes in otherwise wild-type cells. In G1-arrested cells, Mis4 and Rad21 binding to whole nuclei were essentially unchanged by Pef1 ablation (*Figure 4A*). However, their binding to CARs was modified, as revealed by ChIP (*Figure 4B*, *Figure 4—figure supplement 1*). Both Mis4 and Rad21 binding to CARs were increased in a correlated manner in *pef1Δ* cells. The strongest effect was observed at the NTS site (~2.5 fold the wild-type levels), the centromere central core (*cc2*,>1.5 fold) and flanking heterochromatin domains (*imr2-L*, *dg2-R*). Most chromosome arm CARs did respond as well. A non-CAR (ars 3004) site did not show any enrichment for Mis4 and Rad21 (*Figure 4—figure supplement 1*) and did not respond to Pef1 ablation either (*Figure 4B*). These data indicate that Pef1 ablation results in increased binding of cohesin and its loader to CARs without grossly affecting their total amount bound per nucleus.

In cycling cells (which are mainly (80%) in G2 *Carlson et al., 1999*), the effect of Pef1 ablation was less pronounced. Rad21 and Mis4 binding were still enhanced within the centromere, NTS and telomere but the fold increase over wild-type was lower and chromosomal arm sites were essentially unchanged (*Figure 4C*).

Co-immunoprecipitation experiments indicated that the amount of Rad21 bound to Mis4 in G1-arrested cells was enhanced in the absence of Pef1 (*Figure 4D*). The effect was modest (~1.9 fold increase, *Figure 4E*) but consistent between experiments. This may reflect the increased abundance of both cohesin and its loader at CARs. It is of note that Rad21 was hypo-phosphorylated in the absence of the CDK (as detailed below) which may modify how cohesin, Mis4 and DNA interact with each other.

## The Pef1 CDK phosphorylates Rad21

Pef1 may act through the phosphorylation of one or several critical substrates. Pef1 co-immunoprecipitated cohesin and Mis4 from total protein extracts (*Figure 5* AB) and western blot analyses indicated that the phosphorylation state of Rad21 was altered in *pef1* deleted cells (*Figure 5C*). In wild-type, Rad21 displays multiple phospho-isoforms with reduced electrophoretic mobility by SDS-PAGE (*Adachi et al., 2008*; *Birot et al., 2017*; *Birkenbihl and Subramani, 1995* and *Figure 5C*). Fast migrating Rad21 species accumulated in *pef1* deleted cells at the expense of slow migrating forms. To see whether Pef1 directly phosphorylated Rad21, we used in vitro kinase assays. Indeed,

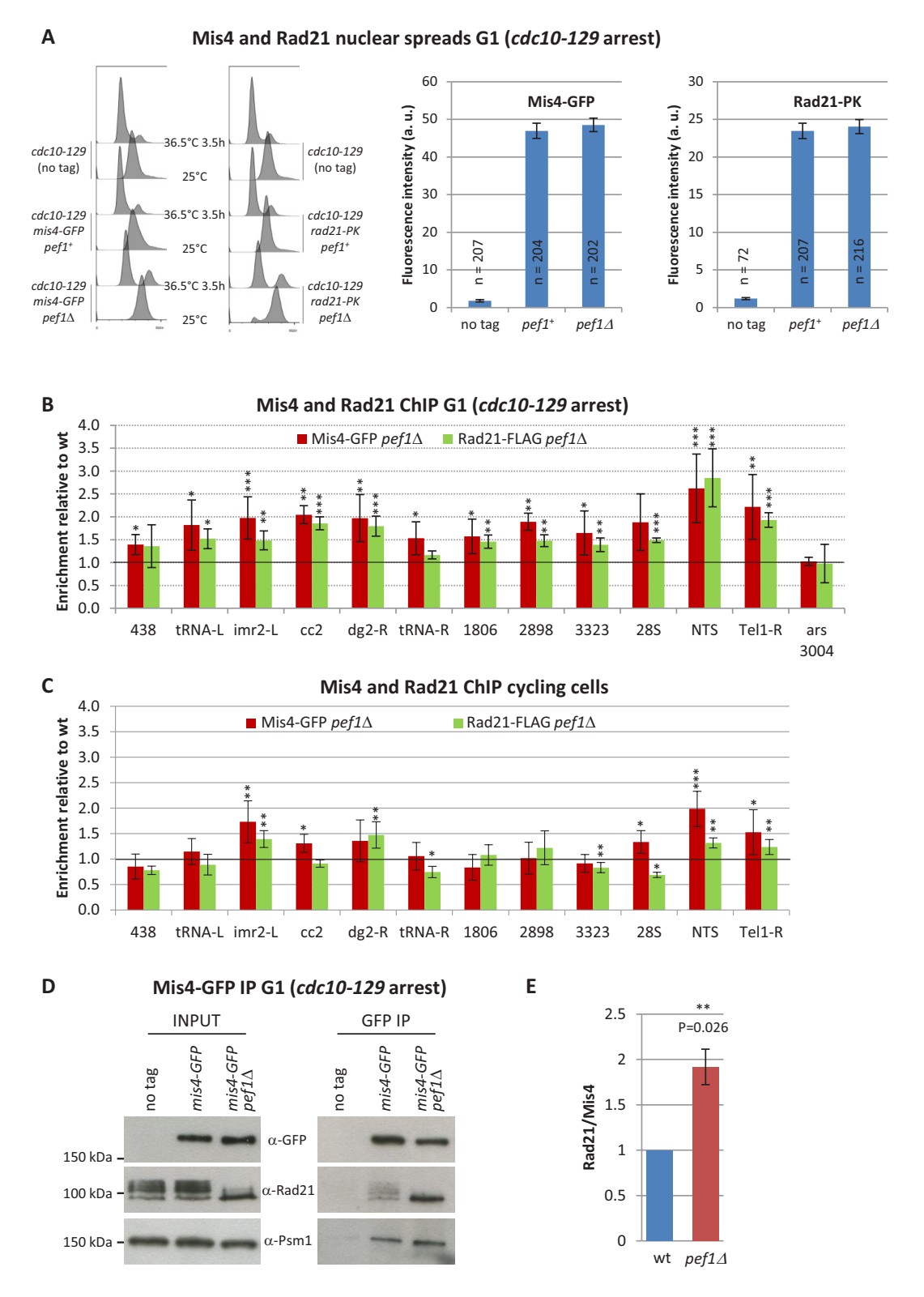

**Figure 4.** Pef1 ablation affects Rad21 and Mis4 binding to G1 chromosomes. (**A**) Mis4-GFP and Rad21-PK binding to whole G1 nuclei. Cells were cultured at 36.5°C to induce the *cdc10-129* arrest and collected after 3.5 hr. G1 arrest was monitored by DNA content analysis. Mis4-GFP and Rad21-PK binding to whole nuclei were measured by nuclear spreads and indirect immunofluorescence. The graphs show the mean fluorescence intensity per nucleus + /- the 95% confidence interval. (**B**) Mis4-GFP and Rad21-FLAG ChIP from G1-arrested cells. Raw ChIP data (*Figure 4—figure supplement 1*)

*Figure 4 continued on next page*

*Figure 4 continued*

were normalized to wild-type levels. Bars indicate mean ± SD, n = 4. A two-tailed, unpaired t-test was used to assess enrichment over wild-type levels. ***p≤0.001, **p≤0.01, *p≤0.05 with 95% confidence interval (*Figure 4—source data 1*). (C) Rad21-FLAG and Mis4-GFP ChIP from cycling cells. ChIP data (*Figure 4—figure supplement 1*) were normalized to wild-type levels. Bars indicate mean ± SD, n = 4. ***p≤0.001, **p≤0.01, *p≤0.05, by two-tailed, unpaired t-test with 95% confidence interval (*Figure 4—source data 1*). (D) Pef1 ablation increased Rad21 binding to its loader. Mis4-GFP was immuno-purified from G1 (*cdc10-129*) protein extracts and co-purifying proteins were analyzed by western blotting with the indicated antibodies. (E) Quantification of Rad21 in Mis4-GFP IPs. Band intensities were measured and the ratios Rad21/Mis4 were normalized to wt. Bar = mean+/-SD from four biological replicates. **p≤0.01 by one sample t test with 95% confidence interval.

The online version of this article includes the following source data and figure supplement(s) for figure 4:

**Source data 1.** Raw ChIP data and t-tests.

**Figure supplement 1.** Mis4-GFP and Rad21-FLAG ChIP from G1 and cycling cells.

the CDK purified from cycling or G1-arrested cells phosphorylated Rad21 (*Figure 5D*). Pef1 was reported to bind three different cyclins. In vitro Rad21 phosphorylation was abolished when Pef1 was purified from *psl1* deleted cells (*Figure 5D*), strongly suggesting that Pef1 acts together with the Psl1 cyclin to phosphorylate Rad21. Accordingly, the electrophoretic mobility of Rad21 was similar in *psl1Δ* and *pef1Δ* cells (*Figure 5C*). To confirm that Pef1 uses the Psl1 cyclin to phosphorylate Rad21, *pef1-GFP* was fused to the endogenous *psl1* gene in a *pef1Δ* background so that Psl1-Pef1-GFP should be the sole Pef1 CDK available in the cell. Indeed the fusion protein purified from cell extracts phosphorylated Rad21 in vitro (*Figure 5E*). In addition, the cohesin core subunit Psm1 efficiently co-purified with Psl1-GFP (*Figure 5F*).

To identify the phosphorylated residue(s), truncated Rad21 peptides were used as substrates for in vitro kinase assays (*Figure 5—figure supplement 1*). A N-terminal fragment (1-356) was efficiently phosphorylated by Pef1 in vitro. Several CDK consensus sites (S/T-P) lie within that region. Replacement of T262 by an alanine abolished in vitro Rad21 phosphorylation by Pef1-GFP (*Figure 5E* and *Figure 5—figure supplement 1*). A weak signal was sometime observed with long exposure times (*Figure 5—figure supplement 1*) or with the Psl1-Pef1 fusion protein (*Figure 5E*), suggesting that other Rad21 residues might be additional or alternative substrates of the kinase. Finally, we raised antibodies against a Rad21-T262 phosphorylated peptide. As shown in *Figure 5G*, the antibodies detected Rad21 purified from wild-type but not from *pef1Δ* cells extracts. From this set of experiment, we conclude that the Pef1/Psl1 CDK phosphorylates Rad21 on threonine 262.

Rad21-T262 phosphorylation was detected in G1, S and G2-arrested cells (*Figure 5—figure supplement 2*). Pef1-as inhibition in G1- and G2-arrested cells lead to a strong decrease of Rad21-T262P, indicating that the CDK was active at these stages of the cell cycle and required for sustained Rad21-T262 phosphorylation. For cells arrested in S phase using hydroxyurea, Rad21-T262P was reduced but not to the same extent, suggesting another kinase may be contributing.

The in vivo relevance of this pathway was assessed by looking at genetic interactions with the cohesin loader mutant *mis4-367*. Individual deletion of the three cyclin genes indicated that *pas1Δ* was a poor *mis4-367* suppressor; the deletion of *clg1* had a weak effect while *psl1Δ* showed the strongest effect (*Figure 5H*). Still, the suppression by *psl1Δ* was weaker than that conferred by *pef1Δ*, suggesting that all three cyclins may act with Pef1 to regulate Mis4 function, likely through the phosphorylation of a set of substrates. Importantly, *rad21-T262A* suppressed the thermosensitive growth defect of *mis4-367* (*Figure 5I*). The level of suppression was similar to *psl1Δ*, consistent with Rad21-T262 being the main relevant substrate of Pef1/Psl1. Conversely, the phospho-mimicking allele *rad21-T262E* exacerbated the ts phenotype of *mis4-367* and compromised the suppression by *psl1Δ*.

From these data, we conclude that all three known Pef1 CDK complexes contribute to cohesin regulation, suggesting multiple relevant substrates. One of these is Rad21. The Pef1/Psl1 CDK phosphorylates Rad21 on residue T262 and genetic analyses suggest that this phosphorylation event may negatively regulate Mis4 function in vivo.

## The phosphorylation status of Rad21-T262 contributes to regulating Mis4 binding to cohesin associated regions

Since Pef1 regulates Mis4 binding to CARs on chromosomes and Rad21 is a substrate of the CDK, we asked whether *rad21-T262A* would recapitulate some aspect of Pef1 loss of function. ChIP

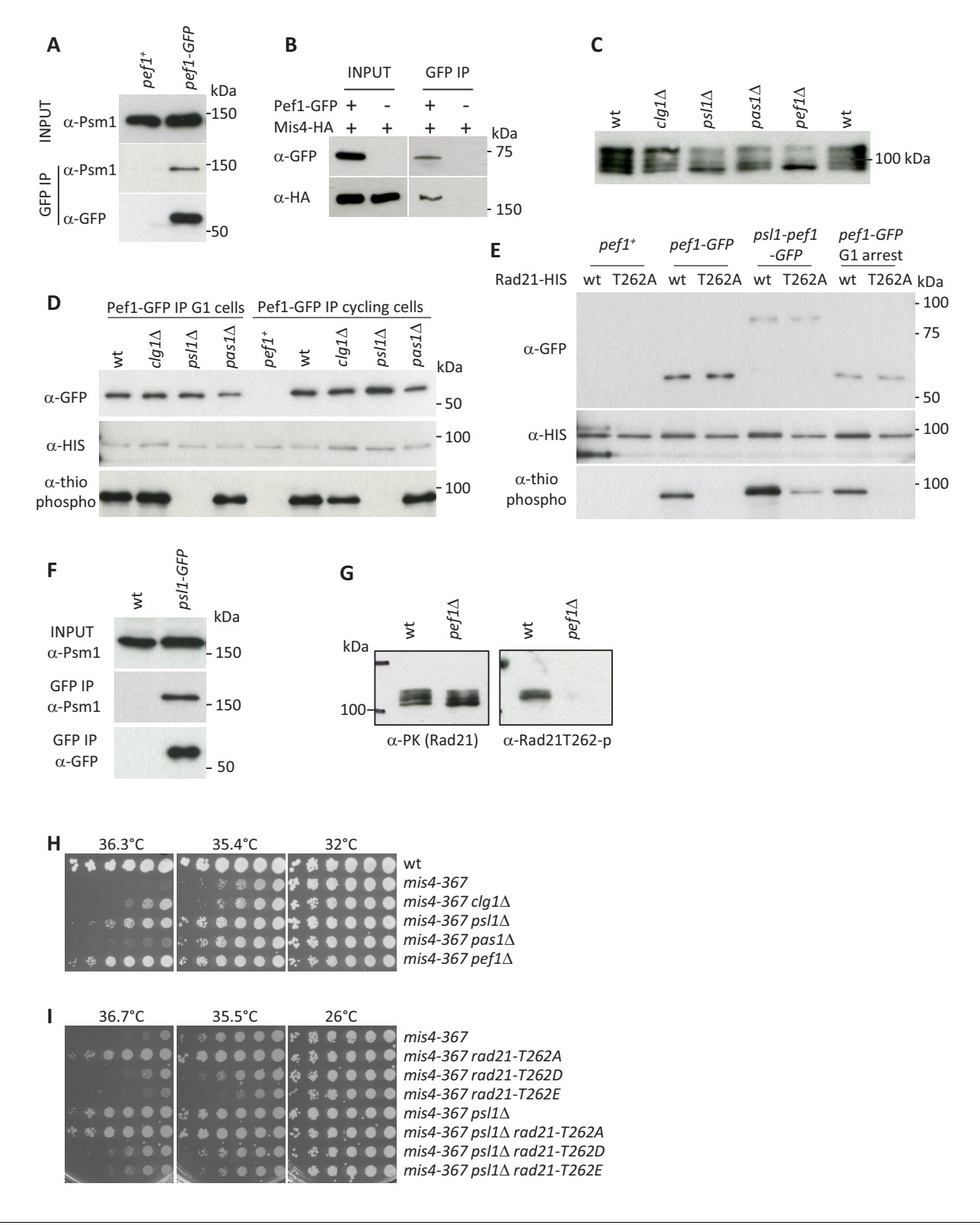

**Figure 5.** Pef1 phosphorylates Rad21. (A,B) Pef1 co-immunoprecipitates cohesin (A) and the cohesin loader Mis4 (B) from total protein extracts. (C) Western blot analysis of total protein extracts from cycling cells probed with anti-Rad21 antibodies. (D) In vitro kinase assays. Pef1-GFP immuno-purified (IP) from cycling or G1 (*cdc10-129*) cells was incubated with in vitro translated Rad21-HIS in the presence of ATPγS and the proteins analyzed by western blotting. Phosphorylated products were detected using an anti-thiophosphate ester antibody. (E) In vitro kinase assays. Rad21-T262A prevents Rad21

*Figure 5 continued on next page*

*Figure 5 continued*
phosphorylation by Pef1. The fusion protein Psl1-Pef1-GFP phosphorylates Rad21. (**F**) Psl1 co-immunoprecipitates Psm1 from total protein extracts (cycling cells). (**G**) Rad21-PK was immuno-purified from cycling cells and probed by western blotting with the indicated antibodies. (**H,I**) Growth assays for suppression of the ts growth defect of *mis4-367*.
The online version of this article includes the following figure supplement(s) for figure 5:

**Figure supplement 1.** Mapping the Pef1 phosphorylation site within Rad21.
**Figure supplement 2.** Rad21-T262 phosphorylation in G1, S and G2-arrested cells.

analyses in G1-arrested cells indicated that Mis4 binding was indeed increased at some loci in a *rad21-T262A* background, although the effect was weaker than for the *pef1* deleted strain (*Figure 6B*). The most prominent effects were observed at the telomere site Tel1-R (1.5 fold increase) and within centromeric heterochromatin (1.4 and 1.5 fold at *imr2-L* and *dg2-R*, respectively). By contrast, Mis4 binding was close to wild-type levels along chromosome arm sites and within the rDNA gene cluster (28S and NTS). No additive effect was seen in combination with *pef1Δ*, consistent with Rad21-T262 being a substrate of the CDK. Mis4 binding was marginally reduced in the phospho-mimicking mutant *rad21-T262E* but importantly, increased Mis4 binding in *pef1Δ* was reduced in a *rad21-T262E* background (*Figure 6C*).

In summary, phosphorylated Rad21-T262 is necessary but not sufficient for Pef1-mediated down-regulation of Mis4 binding. Reciprocally, non-phosphorylatable Rad21-T262 by itself is sufficient to enhance Mis4 binding to specific loci but does not fully recapitulate Pef1 ablation. We conclude that Pef1 regulates Mis4 binding to its chromosomal sites through the phosphorylation of a set of substrates, including Rad21-T262.

## Pef1 and Protein Phosphatase 4 oppose each other

We previously reported that Protein Phosphatase 4 regulated the phosphorylation state of the cohesin subunit Rad21 (*Birot et al., 2017*). Western blot analyses indicate that slow migrating Rad21 isoforms accumulate in a strain deleted for *pph3*, encoding the catalytic subunit of PP4 (*Birot et al., 2017* and *Figure 7A*). Conversely, fast migrating Rad21 isoforms accumulate in a *pef1* deleted strain and a mixed pattern is observed when both the CDK and PP4 are ablated (*Figure 7A*), suggesting that Pef1 and PP4 may oppose each other for controlling the phosphorylation status of Rad21.

A link between PP4 and cohesin loading was provided through the analysis of genetic interactions. Acetyl-mimicking forms of Psm3 are known to inhibit DNA capture by cohesin in vitro and reduce the amount of chromatin-bound cohesin in vivo (*Murayama and Uhlmann, 2015*; *Hu et al., 2015*). A negative genetic interaction was observed between *pph3Δ* and *psm3^{K105NK106N}* (*psm3^{NN}*) an acetyl-mimicking allele of *psm3* (*Feytout et al., 2011*). The double mutant strain was unable to grow at elevated temperature (*Figure 7B*). Interestingly, growth was efficiently rescued by the deletion of *pef1*, *psl1*, and by the non-phosphorylatable allele *rad21-T262A*. The phospho-mimicking allele *rad21-T262E* alone did not recapitulate the effect of deleting *pph3*, suggesting that the negative genetic interaction involves the accumulation of other phosphorylated substrates.

ChIP analyses confirmed that Rad21 binding to chromosomes was reduced in a *psm3^{NN}* background after one complete cell cycle at 36°C at all sites examined (*Figure 7C* and *Figure 7—figure supplement 1*). PP4 ablation exacerbated this phenotype, whereas Pef1 ablation has the opposite effect. Therefore, in a context of compromised cohesin loading (*psm3^{NN}*), PP4 activity stimulated cohesin loading while Pef1 restrained it. Consistent with the genetic data, poor Rad21 binding in the absence of PP4 was efficiently rescued by *pef1Δ* and to a lower extent by *rad21-T262A* (*Figure 7C* and *Figure 7—figure supplement 1*). It is of note that Rad21 binding was slightly higher in *pef1Δ* than in *pef1Δ pph3Δ* at most chromosomal sites, indicating that full stimulation of Rad21 binding by Pef1 ablation requires functional PP4. In the absence of Pef1, some substrates may be phosphorylated by another kinase and de-phosphorylated by PP4. The rDNA gene cluster behaved differently. Rad21 was bound to a similar extent in *pef1* and *pef1 pph3* deleted strains, suggesting that no other kinase phosphorylates Pef1 targets at these loci.

In summary, this experiment revealed that Pef1 and PP4 oppose each other in a situation where the cohesin loading reaction is compromised. Consistently, a similar set of genetic interactions were

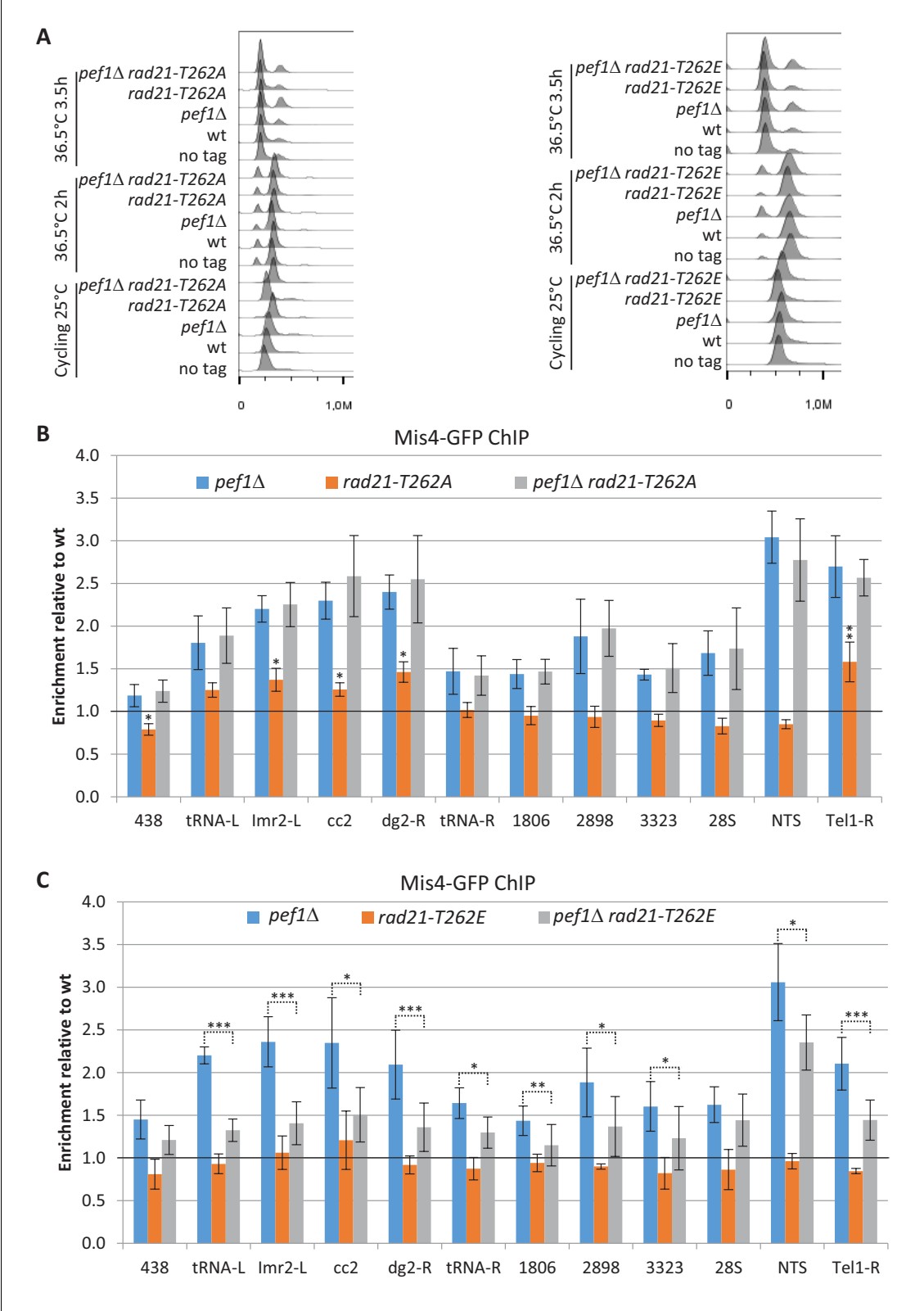

**Figure 6.** Rad21-T262 phosphorylation modulates Mis4 binding to G1 chromosomes. (A) DNA content analysis. Cultures of the indicated strains were shifted to 36.5°C to induce the *cdc10-129* arrest and cells collected for ChIP after 3.5 hr. (B,C) Mis4-GFP ChIP relative to wild-type. Bars indicate mean ± SD, n = 4. ***p≤0.001, **p≤0.01, *p≤0.05, by two-tailed, unpaired t-test with 95% confidence interval (*Figure 6—source data 1*).
*Figure 6 continued on next page*

*Figure 6 continued*

The online version of this article includes the following source data for figure 6:

**Source data 1.** Raw ChIP data and t-tests.

observed when the activity of the cohesin loader was compromised by the *mis4-367* mutation (*Figure 7D*).

To see the effect of PP4 and Pef1 in otherwise wild-type cells, we looked at Rad21 and Mis4 binding to G1 chromosomes by ChIP. PP4 ablation lead to an overall decrease of DNA-bound Rad21 and Mis4 (*Figure 7E*). Conversely, both Rad21 and Mis4 binding were stimulated in the *pef1* deleted strain. The double mutant strain showed a ChIP profile similar although not identical to *pef1Δ* alone. This is consistent with Pef1 and PP4 opposing each other for controlling the phosphorylation state of common substrates: the phosphorylated state (*pph3Δ*) reduces cohesin loading while the non-phosphorylated state (*pef1Δ* and *pef1Δ pph3Δ*) has the opposite effect. The differences observed between *pef1Δ* and *pef1Δ pph3Δ* suggests that other kinase(s) may be contributing.

Co-immunoprecipitation experiments indicated that the amount of Mis4-bound Rad21 was reduced in *pph3* deleted cells and might be restored to some extent when *pef1* was additionally deleted (*Figure 7—figure supplement 1*). Collectively, these data indicate that Pef1 and PP4 oppose each other for regulating the interactions of cohesin with its loader and DNA.

## Rad21 phosphorylation status may modulate the activity of the cohesin loader

The *psm3^NN^ pph3Δ* strain grew poorly and spontaneous suppressors were frequently observed. Genetic analyses showed that the vast majority were allelic to *pef1*. However, five suppressors were allelic to *mis4* and efficiently rescued *psm3^NN^ pph3Δ* growth and chromosome segregation defects (*Figure 8A,D*). Strikingly, all five mutations clustered within the hook domain of Mis4 (*Figure 8B*). Even more striking, the very same mutations were recovered as intragenic suppressors of the ts allele *mis4-G1326E* (*Xu et al., 2018*) suggesting they enhance Mis4 activity. This region is enriched for residues mutated in Cornelia de Lange syndrome and Kikuchi et al. have shown that many of these mutations specifically disrupt the Scc2-Scc1 interaction (*Kikuchi et al., 2016* and *Figure 8C*). In Mis4, this region does not contain any CDK consensus or reported phosphorylation site suggesting it may not be targeted by Pef1/PP4. However, since Rad21 is hyper-phosphorylated in PP4-deprived cells, the suppressor mutations may help Mis4 accommodating a phosphorylated substrate. This possibility is consistent with the finding that *pef1Δ* and *rad21-T262A* mutants are efficient suppressors of *psm3^NN^ pph3Δ*, and that Rad21 phosphorylation was indeed reduced in *pef1Δ pph3Δ* when compared to *pph3Δ* alone (*Figure 7A*). We suggest that the activity of the cohesin loader may be modulated by the phosphorylation of its cohesin substrate.

## Discussion

Cohesin is involved in a wide range of cellular functions at all stages of the cell cycle, implying a tight control by the cell machinery. The data presented here provide evidence that the CDK Pef1 and PP4 are part of this regulatory network. We will discuss here how phosphorylation may control the activity of the cohesin loader and speculate about the potential physiological implications.

In otherwise wild-type cells, Pef1 ablation increased the interaction of both cohesin and its loader Mis4 with their regular binding sites on chromosomes. This concerted increase suggests that cohesin deposition is enhanced because more Mis4 molecules have been recruited to CARs. These cohesin complexes appeared functional as Pef1 inhibition improved sister-chromatid cohesion and chromosome segregation in a *mis4-367* background. Therefore, the most straightforward interpretation is that reduced cohesin loading activity in *mis4-367* is enhanced when the CDK is inactivated, providing sufficient cohesin amenable to sister chromatid cohesion establishment at the time of S phase.

Alternatively, cohesin and its loader may accumulate there as the result of a delay in completing some aspect of Mis4 functions. Besides topological DNA capture cohesin form intra-chromosomal loops which may be generated by a distinct biochemical activity of cohesin, possibly regulated by the loading complex (*Srinivasan et al., 2018*; *Petela et al., 2018*; *Rhodes et al., 2017*). While this

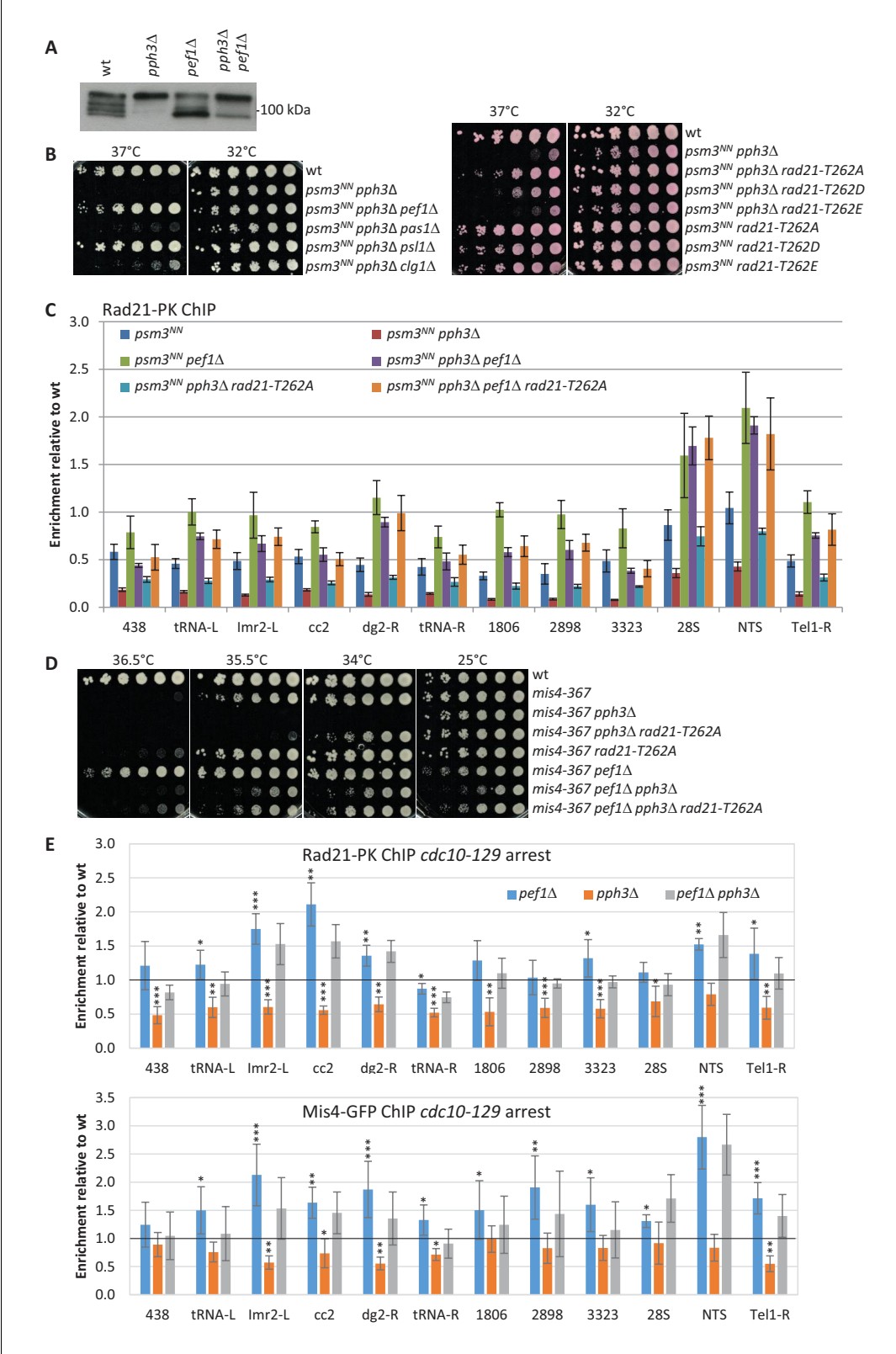

**Figure 7.** Pef1 and PP4 oppose each other for regulating Rad21 binding to chromosomes. (**A**) Western blot analysis of total protein extracts from cycling cells probed with anti-Rad21 antibodies. (**B**) Cell growth assays showing that the ts growth defect of *psm3^NN pph3Δ* is efficiently rescued by *pef1* and *psl1* deletion mutants (left) and *rad21-T262A* (right). (**C**) Rad21-ChIP after one complete cell cycle at 36°C. Data are expressed relative to wild-type. Bars indicate mean ± SD from four ratios. t-tests are shown in *Figure 7—figure supplement 1* and *Figure 7—source data 1*. (**D**) Cell growth

*Figure 7 continued on next page*

*Figure 7 continued*

assays showing that *pef1Δ*, *rad21-T262A* and *pph3Δ* display opposite genetic interactions with *mis4-367*. (E) Rad21 and Mis4 ChIP relative to wild-type in *cdc10-129*-arrested cells. Bars indicate mean ± SD from four ratios (*Figure 4—source data 1*).

The online version of this article includes the following source data and figure supplement(s) for figure 7:

**Source data 1.** Raw ChIP data and t-tests.
**Figure supplement 1.** Pairwise comparisons of ChIP data from *Figure 7C* and quantification of Rad21 in Mis4 immunoprecipitates.

manuscript was in revision two new studies were published, demonstrating in vitro DNA loop extrusion by human cohesin (*Davidson et al., 2019*; *Kim et al., 2019*). Crucially, loop formation and maintenance depend on the cohesin loader which, together with cohesin, forms an active holoenzyme residing at the base of loops. Pef1 may positively regulate the formation of loops. In this scenario, inhibition of the CDK may rescue sister-chromatid cohesion in Mis4 compromised cells by increasing the pool of cohesin available for cohesion at the expense of those engaged in DNA looping. Further studies will address this attractive possibility.

What would be the physiological roles of a CDK-based regulation of cohesin? As mentioned above, the CDK may regulate non-cohesive aspects of cohesin functions with consequences on gene expression and nuclear architecture. The most prominent effect on Mis4 and Rad21 binding to chromosomes were observed in G1, a cell cycle stage where Pef1 has a role in facilitating cell cycle progression (*Tanaka and Okayama, 2000*). G1 is the critical stage where cells integrate signals from their environment and decide to embark into another cell cycle, enter quiescence or differentiate. The Pef1 control of cohesin behavior we describe here may be part of the cell's response to input signals conveyed by the CDK. The nature of the input signals and the physiological outputs are important issues to address in the future.

Pef1 may also play roles outside the G1 phase as the CDK did phosphorylate Rad21-T262 in cells arrested in S and G2 phases. Pef1 activity during S phase may play a role in the establishment of sister chromatid cohesion. Pef1 ablation in otherwise wild-type cells did not lead to obvious chromosome segregation defects, suggesting no adverse effect on the establishment of sister chromatid cohesion. However, *pef1* mutants showed a negative genetic interaction with *eso1-H17* which is deficient for cohesin acetylation (*Feytout et al., 2011*), suggesting a positive role for Pef1 in sister chromatid cohesion establishment or maintenance that would remain cryptic when Eso1 is fully functional. Finally, a CDK-based control of cohesin may be relevant to DNA damage response. In support of these ideas, the closest Pef1 human homolog, CDK5, has been implicated in DNA damage response and gene regulation (*Liu et al., 2017*).

Understanding how Pef1 regulates cohesin binding to DNA will require further knowledge of the biochemical activities of both cohesin and its loader. Mis4/Ssl3 interacts with all cohesin subunits and these contacts contribute to the activity of the cohesin loader (*Murayama and Uhlmann, 2014*). Structural studies revealed a high conformational flexibility of Scc2 suggesting that the loader may capture cohesin by making multiple contacts around the surface of the ring that may help conformational changes required for DNA capture (*Chao et al., 2015*). Of particular interest for the present study is the interaction between the loader and Rad21. We identified Rad21-T262 as a Pef1 substrate which when phosphorylated contributes to down-regulating cohesin binding to CARs. Threonine 262 is located within the central, unstructured domain of Rad21. Interestingly, Rad21-T262 lies between the two Mis4/Ssl3 contact sites (145–152 and 408–422) that were mapped on Rad21 by peptide arrays (*Murayama and Uhlmann, 2014*) and adjacent to the Scc2 binding site (126–230) on *Chaetomium thermophilum* Scc1 (*Kikuchi et al., 2016*). Rad21 phosphorylation may therefore hinder or modify Mis4 interaction with cohesin. Our co-immunoprecipitation assay is consistent with this possibility as the interaction between Mis4 and cohesin appeared slightly increased in the absence of Pef1. Although the central domain of Rad21 is poorly conserved across species, numerous phosphorylation sites have been mapped within that region in human Rad21 (http://www.phosphosite.org/), suggesting that a similar regulatory mechanism may operate in human. Another argument came from the suppressors of the negative interaction between *psm3^NN* and *pph3Δ*. Besides *pef1* mutants and *rad21-T262A*, suppressor mutations were found in Mis4 and clustered within a Rad21-binding domain (*Kikuchi et al., 2016*). These amino acid changes may help accommodate hyperphosphorylated Rad21 when PP4 is ablated.

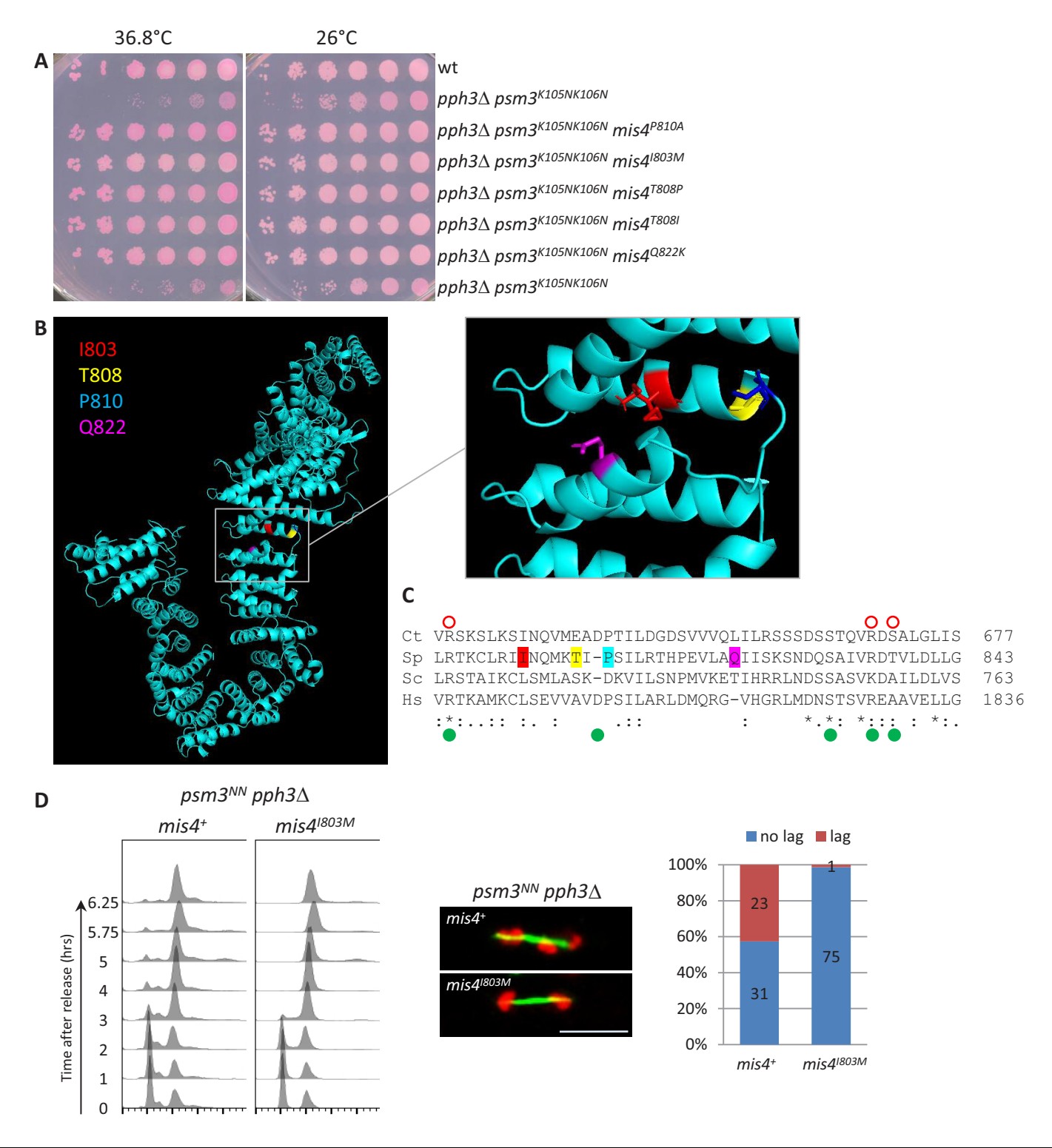

**Figure 8.** Mis4[I803M] suppresses *pph3Δ psm3[NN]* chromosome segregation defects. (**A**) Cell growth assay. The *mis4* mutations restore *pph3Δ psm3[NN]* growth at elevated temperature (**B**) Model structure of Mis4 showing the location of the mutated residues. (**C**) Sequence alignment of Mis4 with Scc2 proteins from other species. The mutated residues in Mis4 are colored as in (**B**). Ct Scc2 residues required for Scc1 binding are indicated by open red circles and Hs Scc2 residues mutated in Cornelia de Lange syndrome are indicated by green dots as in *Kikuchi et al. (2016)*. Ct, *Chaetomium thermophilum*; Sc, *Saccharomyces cerevisiae*; Sp, *Schizosaccharomyces pombe*, Hs, *Homo sapiens*. (**D**) Mis4[I803M] efficiently suppressed *pph3Δ psm3[NN]* chromosome segregation defects. Cells were arrested in G1 by nitrogen starvation at 25°C and released at 37°C. Progression in the cell cycle was

*Figure 8 continued on next page*

*Figure 8 continued*
monitored by DNA content analysis. Cells from the 5.75 and 6.25 time points were processed for DNA and tubulin staining to score the number of anaphase cells with lagging chromatids. Bar = 5µm.

The negative charges bring about by phosphorylated residues may also hinder or modify cohesin interaction with DNA. A recent study indicates that cohesin would tether DNA in its smaller lumen that is, between Rad21 and the SMC's heads (*Chapard et al., 2019*). The two acetylatable lysine residues within Psm3 head domain are thought to stimulate the ATPase activity of cohesin when in contact with DNA (*Murayama and Uhlmann, 2014*). Rad21 phospho-residues may alter the path of the DNA along Rad21 and hinder its contact with Psm3 lysine sensors. Alternatively or additionally, Rad21 phosphorylation may affect the recently reported DNA binding interface between Scc1 and Scc3, the respective budding yeast counterparts of Rad21 and Psc3 (*Li et al., 2018*). In essence, hypo-phosphorylated cohesin may favor the interaction with either of both the cohesin loader and DNA, and reciprocally, targeted phosphorylation events may have the opposite issue. A phosphorylation-based control of cohesin is appealing as these modifications are reversible and occur within seconds with high spatial resolution. The CDK/PP4 module may provide a fast and accurate mechanism for translating cellular cues into an appropriate cohesin response. We have shown here that Pef1 ablation rescued sister-chromatid cohesion defects of a crippled cohesin loader. Such a regulation may impinge on the other functions of cohesin such as DNA repair and intra-chromosomal looping. Considering the conservation of the Pef1 CDK and PP4 across species, a similar regulation may operate in larger eukaryotes, including humans.

## Materials and methods

### Strains, media and genetic techniques

General fission yeast methods, reagents and media are described in *Moreno et al. (1991)*. All strains are listed in *Supplementary file 1*: Key Resources Table. Experiments were carried out using YES medium unless otherwise stated. Gene deletions and epitope tagging were performed by gene targeting using polymerase chain reaction (PCR) products (*Bähler et al., 1998*). The strain carrying an ectopic copy of *rad21-FLAG* was constructed by integrating a *tetO7-rad21-FLAG* construct into a gene free region on chromosome 3 (*Fennessy et al., 2014*). The tetracycline sensitive repressor was introduced by crossing with *ura4$^+$-tet$^{ON}$* (*tetR-tup11D70* integrated at the *ura4* locus *Zilio et al., 2012*). Expression was induced by the addition of 5 µg/ml tetracycline (anydrotetracycline hydrochloride, SIGMA, stock solution 10 mg/ml in DMSO) or DMSO alone for the un-induced control.

Suppressors of the *mis4-367* thermosensitive growth phenotype were obtained either with or without UV mutagenesis. Cells were plated onto YES medium, irradiated with UV light to ~50% killing, and incubated at 25°C until colony formation. For spontaneous suppressors, the UV irradiation step was omitted. Colonies were replica plated onto YES plates containing the vital dye Phloxin B and incubated over night at 37°C. Suppressors appeared as white growing colonies in an otherwise background of red-stained dead cells. Suppressors were backcrossed at least three times. Eleven suppressors showed a monogenic segregation and fell into four linkage groups. Genetic mapping of group one mutants indicated linkage with the *cds1* and *sds21* loci on chromosome III. The mutated locus was identified by Comparative Genome Hybridization (CGH). Genomic DNA was extracted from the mutant strain *sup$^{UV12}$* and the wild-type *S. pombe* reference strain (SP972) and co-hybridized to a CGH tiling array (29–32 mer probes with seven or eight base spacing from the start of one probe to the start of the next, Roche Nimblegen). The array spanned 1,290,000 bp of chromosome 3, from coordinates 117000 to 1065000 and from 1143000 to 1485000 (Genbank NC_003421.2 GI:63054406). DNA regions carrying candidate Single Nucleotide Polymorphisms (SNPs) were used to design a high-resolution tiling array (29–30 mer probes tiled such as each candidate SNP is analyzed by eight probes, four on each DNA strand). A single A to G SNP (N146S) was found within SPCC16C4.11 (*pef1*). The mutation was confirmed by PCR and DNA sequencing. The *pef1* gene was deleted and genetic analyses showed that *pef1Δ* was allelic to *sup$^{UV12}$* and suppressed the Ts phenotype of *mis4-367*.

The *pef1-as* allele (*pef1-F78A*) was generated by in vitro mutagenesis. The DNA fragment carrying the mutated allele was transformed into a recipient strain in which the region of interest was substituted by the *ura4$^+$* marker (*pef1d(260-320)::ura4$^+$*), and Ura$^-$ colonies were selected on 5-Fluoroorotic acid plates. The *pef1-as* allele in the selected strain was amplified by PCR and checked by sequencing. The inhibitor 1-NA-PP1 (Cayman Chemical, stock solution 25 mM in DMSO) was added to the culture medium at 25 µM. The equivalent volume of solvent alone (DMSO) was added for the control sample.

The construction of *rad21* alleles was done using the strategy described in *Birot et al. (2017)*.

## Cytological techniques

DNA content was measured by flow cytometry with an Accuri C6 Flow cytometer after Sytox Green staining of ethanol-fixed cells (*Knutsen et al., 2011*). Data were presented using the FlowJo software. Immunofluorescence and Fluorescence In Situ Hybridization (FISH) were done as described (*Steglich et al., 2015*). Briefly, cells were fixed by the addition of paraformaldehyde to a final concentration of 1.8% in 1.2M sorbitol. The flasks were removed from 36°C, incubated at 21°C for 45 min and processed for tubulin staining using TAT1 antibodies (*Woods et al., 1989*). Cells were refixed and processed for FISH using the centromere linked c1228 cosmid as a probe (*Mizukami et al., 1993*). Cells were imaged using a Leica DMRXA microscope and a 100X objective. Tubulin staining was used to select cells with an interphase array of microtubules. Distances between FISH signals were measured from maximum projections of images created from z series of eight 0.4 µm steps using MetaMorph software. Cen2FISH signals were considered as separated when the distance was greater than 0.3 µm. Statistical analysis was done using two-tailed Fisher exact test with 95% confidence interval using the GraphPad software. Nuclear spreads were done as described (*Feytout et al., 2011*). Signal intensity was measured in a square surface containing the spread nucleus. Background signal was measured by moving the square surface to an adjacent region devoid of nuclei. The background value was subtracted for each nucleus. The signal was quantified for at least 35 nuclei for each sample. The mean and the confidence interval of the mean were calculated with $\alpha = 0.05$.

## Antibodies, protein extracts, immunoprecipitation, western blotting, cell fractionation, chromatin immunoprecipitation (ChIP) and kinase assay

Rabbit polyclonal antibodies against Rad21, Psm1, Psm3, Psm3-K106Ac have been described previously (*Feytout et al., 2011*; *Dheur et al., 2011*). The mouse monoclonal anti-tubulin antibody TAT1 is from *Woods et al. (1989)*. Anti-Rad21-T262P antibodies were raised by Biotem (Apprieu, France). Rabbits were immunized with the KLH-coupled peptide C+SVTHFSTpPSMLP. Sera were immune-depleted by affinity with the non-phosphorylated form of the peptide and antibodies were affinity purified against the phosphorylated peptide. Other antibodies were of commercial source (*Supplementary file 1*: Key Resources Table). Protein extracts, immunoprecipitation (IP), cell fractionation and western blotting were as described (*Feytout et al., 2011*; *Schmidt et al., 2009*). Quantification of western blots was done using ImageJ. Chromatin Immunoprecipitation (ChIP) was as described in *Birot et al. (2017)* using anti-FLAG, anti-PK or anti-GFP (A11122) antibodies. ChIP enrichments were calculated as percentage of DNA immunoprecipitated at the locus of interest relative to the input sample. The mean was calculated from four technical replicates with error bars representing standard deviation. Enrichments relative to wild-type were calculated as the mean of four ratios with errors bars representing standard deviation. Statistical analyses (t-tests) were done using the GraphPad software. qPCR primers are listed in *Supplementary file 1*: Key Resources Table.

For kinase assays, the Rad21-6HIS substrates were produced using a coupled transcription/translation reaction system (*E. coli* EasyXpress, biotechrabbit) using plasmid DNA templates. The reaction was carried out at 37°C for 1 hr using 5–10 nM plasmid DNA in a total volume of 50 µL. Pef1-GFP was immunoprecipitated (IPed) from total cell extracts ($5 \times 10^8$ cells) prepared in lysis buffer (50 mM Hepes pH 7.6; 75 mM KCl; 1 mM MgCl2; 1 mM EGTA; 0.1% Triton X-100; 1 mM DTT; 10 mM Sodium butyrate; Glycerol 10%) supplemented with inhibitors (Protease inhibitor cocktail Sigma P8215, 1 mM PMSF, 1 mM Na vanadate, 20 mM β-glycerophosphate). The IPed material was washed three times with 0.2 ml of lysing buffer without inhibitors, and twice with 1.5X kinase buffer (75 mM

TRIS pH 7.5; 15 mM MgCl2; 1.5 mM EGTA; 1.5 mM DTT). The CDK bound to the beads was recovered in 60 µL of the 1.5X kinase buffer. The kinase assay was done using 40 µL of CDK beads and 20 µL of in vitro generated Rad21-6HIS using the nonradioactive method described in *Allen et al. (2007)*. The reaction was carried out with 1 mM ATPγS at 37°C for 1 hr with 1 min shaking (300 rpm) every 10 min. The reaction was stopped by adding EDTA to 20 mM. To alkylate thio-phosphorylated proteins p-nitrobenzyl mesylate (Abcam 138910) was added to 5 mM and the samples incubated for 2 hr at 21°C on a rotating wheel. The beads were removed by loading the sample on a magnetic column equilibrated with kinase buffer. The flow-through was collected and the presence of thio-phosphorylated proteins was assayed by western blotting with anti-thioester antibodies.

## Acknowledgements

We thank our colleagues M-F Giraud and S Manon for their initial help and advice for the in vitro production of proteins and kinase assays, K Gull for the gift of anti-tubulin antibodies, K Tanaka and H Okayama from providing *pef1* and *pas1* deleted strains. This work was supported by the Centre National de la Recherche Scientifique, l'Université de Bordeaux, la Région Aquitaine, l'Association pour la Recherche sur le Cancer (PJA 2013 1200 205 ; PJA 20171206211) and l'Agence Nationale de la Recherche (ANR-14-CE10-0020-01). Adrien Birot was supported by a fellowship from the Agence Nationale de la Recherche Investissements d'Avenir ANR-10-IDEX-03–02 and l'Association pour la Recherche sur le Cancer (DOC20160603884). Amélie Feytout was supported by a fellowship from the Ministère de l'Enseignement Supérieur et de la Recherche. Karl Ekwall was supported by grants from the Swedish Cancer Society (CF) and the Swedish Research Council (VR).

## Additional information

### Funding

| Funder | Grant reference number | Author |
|---|---|---|
| Fondation ARC pour la Recherche sur le Cancer | PJA 2013 1200 205 | Jean-Paul Javerzat |
| Fondation ARC pour la Recherche sur le Cancer | PJA 20171206211 | Jean-Paul Javerzat |
| Agence Nationale de la Recherche | ANR-14-CE10-0020-01 | Jean-Paul Javerzat |
| Agence Nationale de la Recherche | ANR-10-IDEX-03-02 | Adrien Birot |
| Fondation ARC pour la Recherche sur le Cancer | DOC20160603884 | Adrien Birot |
| Ministère de l'Enseignement Supérieur et de la Recherche Scientifique | | Amélie Feytout |
| Swedish Cancer Society | | Karl Ekwall |
| Swedish Research Council | | Karl Ekwall |

The funders had no role in study design, data collection and interpretation, or the decision to submit the work for publication.

### Author contributions

Adrien Birot, Marta Tormos-Pérez, Sabine Vaur, Conceptualization, Formal analysis, Investigation, Methodology, Writing - review and editing; Amélie Feytout, Julien Jaegy, Dácil Alonso Gil, Investigation, Writing - review and editing; Stéphanie Vazquez, Investigation; Karl Ekwall, Conceptualization, Funding acquisition, Writing - review and editing; Jean-Paul Javerzat, Conceptualization, Formal analysis, Supervision, Funding acquisition, Validation, Investigation, Visualization, Methodology, Writing - original draft, Project administration, Writing - review and editing

**Author ORCIDs**

Jean-Paul Javerzat (iD) https://orcid.org/0000-0002-9671-6753

**Decision letter and Author response**

Decision letter https://doi.org/10.7554/eLife.50556.sa1
Author response https://doi.org/10.7554/eLife.50556.sa2

## Additional files

### Supplementary files

• Supplementary file 1. Key Resources Table.

### Data availability

All data generated or analysed during this study are included in the manuscript and supporting files. Source data files have been provided for Figure 1, 2, 4, 6 and 7.

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
