## [Decision Letter]

**Acceptance summary:**

The organization of DNA in the nucleus by cohesion, either by formation of DNA loops or by tethering sister chromatids after DNA replication is increasingly being recognized as a key regulator of many nuclear events, including chromatin organization and gene transcription. The regulation of cohesion is complex, but critical to understand in the many contexts of cohesion biochemistry. The current paper provides new data on the opposing roles of a Cyclin-dependent protein kinase Pef1 and Protein Phosphatase 4 (PP4) in cohesion regulation, via phosphorylation of the Rad21 cohesion subunit, thereby affecting the activity of the Mis4 cohesion loader.

**Decision letter after peer review:**

Thank you for submitting your article "The CDK Pef1 and Protein Phosphatase 4 oppose each other for regulating cohesin binding to fission yeast chromosomes" for consideration by *eLife*. Your article has been reviewed by three peer reviewers, including Bruce Stillman as the Reviewing Editor and Reviewer #3, and the evaluation has been overseen by James Manley as the Senior Editor.

The reviewers have discussed the reviews with one another and the Reviewing Editor has drafted this decision to help you prepare a revised submission.

Summary:

In this manuscript, the authors use a genetic approach to identify a previously unknown level of posttranslational regulation of the chromosomal cohesin complex. Phosphorylation by the Pef1-CDK limits cohesin loading and/or stability on chromosomes while the counteracting PP4 phosphatase facilitates loading and/or increases the stability of cohesin binding. The authors identify one of the sites that is targeted by this mechanism as Rad21-T262. This is important information, if a comprehensive understanding of how cohesin is regulated on chromosomes is sought. The experiments are generally conducted and documented well and they support the authors conclusions. As such revisions are requested, taking into account the comments below.

Major comment:

1) The main concern relates to the physiological significance of the mode of regulation depicted in this manuscript. The effect on chromosomal cohesin levels are small, invisible by cytological means. Previously, the authors reported that the PP4 phosphatase destabilizes cohesin on chromosomes (Birot et al., 2017), the opposite of what is reported here. Stabilization in that case was achieved by Rad21 phosphorylation on positions S163/S164 by an unknown kinase. No doubt, the phospho-regulation of Rad21 is intricate. Nevertheless, as it stands, it remains unclear whether either S163/S164 or T262 phosphorylation is used to modulate cohesin behavior in response to any regulatory input. Some discussion on this point is requested.

---

## [Author Response]

Major comment:1) The main concern relates to the physiological significance of the mode of regulation depicted in this manuscript. The effect on chromosomal cohesin levels are small, invisible by cytological means. Previously, the authors reported that the PP4 phosphatase destabilizes cohesin on chromosomes (Birot et al., 2017), the opposite of what is reported here. Stabilization in that case was achieved by Rad21 phosphorylation on positions S163/S164 by an unknown kinase. No doubt, the phospho-regulation of Rad21 is intricate. Nevertheless, as it stands, it remains unclear whether either S163/S164 or T262 phosphorylation is used to modulate cohesin behavior in response to any regulatory input. Some discussion on this point is requested.

Our study reports a previously unknown level of cohesin regulation by a pair of cell cycle kinase/phosphatase. The identity of the input signals and the physiological outputs are indeed very exciting questions but these are still unanswered and form the basis of our current and future projects. As suggested we have made additional experiments that provide some additional information and have remodeled and expanded the Discussion section in that direction.